Beech cupules as keystone structures for soil fauna

Melguizo-Ruiz Nereida nereidamelguizoruiz@gmail.com 1
Jiménez-Navarro Gerardo 2
Moya-Laraño Jordi 1
1 Functional and Evolutionary Ecology, Estación Experimental de Zonas Áridas—CSIC , Almería , Spain
2 CIBIO/InBio-UE Research Center in Biodiversity and Genetic Resources, University of Évora , Évora , Portugal
Sanders Nathan
Electronic publication date: 2016 Oct 19
Publication date: 2016
Volume: 4
Electronic Location ID: e2562
Received 2016 Jun 2; Accepted 2016 Sep 13
Copyright: ©2016 Melguizo-Ruiz et al.
Copyright year: 2016
Copyright holder: Melguizo-Ruiz et al.
License: This is an open access article distributed under the terms of the Creative Commons Attribution License, which permits unrestricted use, distribution, reproduction and adaptation in any medium and for any purpose provided that it is properly attributed. For attribution, the original author(s), title, publication source (PeerJ) and either DOI or URL of the article must be cited.
License URL: https://creativecommons.org/licenses/by/4.0/

Keywords: Soil fauna, Keystone structure, Moisture, Above-belowground linkages, Fagus sylvatica, Beech cupule, Facilitation

Funding: Spanish Ministry of Science and Innovation CGL2010-18602 CGL2015-66192-R European Regional Development Fund FPI fellowship BES-2011-043505 This work has been funded by Spanish Ministry of Science and Innovation grant CGL2010-18602 and CGL2015-66192-R to JML, the European Regional Development Fund and FPI fellowship (BES-2011-043505) to NMR. The funders had no role in study design, data collection and analysis, decision to publish, or preparation of the manuscript.

==============================
Facilitative or positive interactions are ubiquitous in nature and play a fundamental role in the configuration of ecological communities. In particular, habitat modification and niche construction, in which one organism locally modifies abiotic conditions and favours other organisms by buffering the effects of adverse environmental factors, are among the most relevant facilitative interactions. In line with this, ‘keystone structures’, which provide resources, refuge, or advantageous services decisive for other species, may allow the coexistence of various species and thus considerably contribute to diversity maintenance. Beech cupules are woody husks harbouring beech fruits that remain in the forest soil for relatively long periods of time. In this study, we explored the potential role of these cupules in the distribution and maintenance of the soil fauna inhabiting the leaf litter layer. We experimentally manipulated cupule availability and soil moisture in the field to determine if such structures are limiting and can provide moist shelter to soil animals during drought periods, contributing to minimize desiccation risks. We measured invertebrate abundances inside relative to outside the cupules, total abundances in the leaf litter and animal body sizes, in both dry and wet experimental plots. We found that these structures are preferentially used by the most abundant groups of smaller soil animals—springtails, mites and enchytraeids—during droughts. Moreover, beech cupules can be limiting, as an increase in use was found with higher cupule densities, and are important resources for many small soil invertebrates, driving the spatial structure of the soil community and promoting higher densities in the leaf litter, probably through an increase in habitat heterogeneity. We propose that fruit woody structures should be considered ‘keystone structures’ that contribute to soil community maintenance. Therefore, beech trees may indirectly facilitate soil fauna activities through their decaying fruit husks, hence acting as ecosystem engineers.

Introduction

Species can influence each other through both negative and positive interactions (Jones, Lawron & Shachak, 1997; Callaway et al., 2002), although ecological research on the latter is still underrepresented (Bertness & Callaway, 1994; Bronstein, 1994; Bruno & Bertness, 2001; Bruno, Stachowicz & Bertness, 2003; Brooker et al., 2008). Such positive or facilitative interactions, defined as non-trophic interspecific interactions characterized by the benefit of at least one of the partners while negatively affecting neither (Bertness & Callaway, 1994; Bruno, Stachowicz & Bertness, 2003), are ubiquitous in nature and of crucial importance in the configuration of ecological communities (Bertness & Leonard, 1997; Bruno, Stachowicz & Bertness, 2003; Cavieres & Badano, 2009). In fact, diverse studies have empirically shown how facilitation can strongly impact individual fitness, population structure and dynamics, species abundance and diversity as well as landscape-scale community dynamics (Callaway, 1995; Bertness & Leonard, 1997; Jones, Lawron & Shachak, 1997; Bruno & Bertness, 2001; Stachowicz, 2001; Cebrian & Uriz, 2006; Palmer et al., 2013). Some of the most determinant facilitative interactions are related to habitat modification and niche construction, in which one organism locally alters abiotic conditions and subsequently favours other organisms by buffering adverse or stressful habitat conditions (Bruno & Bertness, 2001; Stachowicz, 2001; Crain & Bertness, 2006). These habitat alterations can hence make the local environment more suitable for the maintenance of other members of the community, and potentially contribute to its stability (Stachowicz, 2001). In fact, inclusion of positive or facilitative interactions can change how we understand the niche concept, as it can modify the link between the fundamental and realized niches, making the latter broader than previously expected (Bruno, Stachowicz & Bertness, 2003). In line with this, Tews et al. (2004) proposed the ‘keystone structure’ concept, defined as a distinct spatial structure that provides resources, refuge, or supply beneficial services, critical for other species. Keystone structures, detectable on a particular spatial scale, bring along crucial ecological conditions to a large proportion of species, and therefore may substantially contribute to diversity maintenance (Tews et al., 2004). For instance, female trees of several species produce nuts shielded by woody structures, such as acorns or cupules, that remain in the forest soil together with other dead tree tissues for relatively long periods of time. These husks, that originate as reproductive structures from the aboveground, can potentially play a significant role in the distribution of soil fauna, given that some of them are effective empty cavities which can be used as shelters. Specifically in beech forests, cupules (Appendix 1, Fig. S1) are quantitatively important elements that are the second largest component of the forest litter, and can constitute up to one third of the total annual litterfall in masting years, as documented in a Japanese beech forest of Fagus crenata (Kawada & Maruyama, 1986, cited in Fukasawa et al., 2012 p. 735). If these woody organs can alter the soil physical structure, create additional habitats or niches and provide refuge thereby favouring soil animals, they could be considered keystone structures in the leaf litter. Further, trees could act as ecosystem engineers through their reproductive structures, by influencing soil community structure and its associated ecosystem processes.

Studies focusing on fruit husks and nuts have centred the attention on fungal succession during their decay (Carré, 1964; Fukasawa et al., 2012), insect pests and seed predation by insects and vertebrates (Nilsson & Wästljung, 1987; Gribko, 1995; Dunning, Paine & Redak, 2002; Dalgleish, Shukle & Swihart, 2012; Yu et al., 2015) or tree seeding as a factor affecting the abundance of small mammals and some litter-dwelling arthropods (Fitzgerald et al., 1996; Alley et al., 2001). Most of the research has thus focused on the role that these acorns or cupules play in tree recruitment and population dynamics, while few studies have addressed the question of their contribution to the structure and dynamics of invertebrate food webs, nor on the diversity and abundance of organisms they harbour.

Winston (1956) conducted a pioneering study in which he examined the animals and fungi living in Quercus rubra acorns, and listed all the organisms observed inside the nuts. Among others, numerous species of nematodes, annelids, centipedes, arachnids and insects were detected and assigned to feeding guilds (Winston, 1956). In fact, many invertebrates colonize fruit husks, either momentarily or during larger periods. For instance, some species of mites (Phthiracarids) use beech cupules to lay their eggs, and their endophagous stages can live within the woody tissues for months (Harding & Easton, 1984). Indeed, fruit husks could certainly play an important role in the structure and dynamics of the soil food web, as large amounts of animals are found inside (Winston, 1956; N Melguizo-Ruiz, pers. obs., 2012). Beech cupules are probably a valuable resource for xylophagous and fungivorous organisms, given their ligneous composition and that they are commonly colonized by several species of fungi (Fukasawa et al., 2012). Moreover, these woody husks may contribute to increase habitat heterogeneity and provide shelter to invertebrate species, either from predation–as a hiding place–or to prevent desiccation—given that most soil animals need relatively high soil humidity to minimize water loss (Edney, 1977; Wolters & Ekschmitt, 1997), and therefore any sheltering structure favouring moisture maintenance may probably be used by soil fauna during drought periods. Low relative humidity and changes in precipitation are, in general, stressful for most invertebrate species (Hawkins & Holyoak, 1998; Branson, 2008; Chown, Sørensen & Terblanche, 2011), which may have to seek for shelter or damp refuges during droughts, in order to avoid desiccation (Wolters & Ekschmitt, 1997). Smaller animals can be even more vulnerable to desiccation and lose relatively more water during droughts, because of the higher surface to volume ratio imposed by their small body mass (Chown, 1993; Le Lagadec, Chown & Scholtz, 1998; Renault & Coray, 2004).

Here we aimed to determine the potential effect that cupules can have on the micro-distribution of soil animals inhabiting the leaf litter layer of beech forests, under either wet or dry conditions. We experimentally tested for potential invertebrate preferences for the cupules as a form of micro-habitat selection within the leaf litter layer, especially under dry conditions. This was a first step to understand if beech cupules can provide wet shelter to soil animals, contributing to minimize desiccation risks during drought periods. In a split-plot design, we manipulated water availability (main plots) and then the amount of cupules (subplots) in the leaf litter, and we tested for differences in animal numbers found inside relative to outside the cupules (micro-habitat selection), the spatial distribution of the total abundance of fauna (inside + outside), as well as for differences in animal body masses inside vs. outside cupules in relation to water availability. We hypothesized that (1) water availability would positively influence the micro-distribution of soil fauna (i.e., at the plot level), with an increase in invertebrate activity and abundance in wet plots, probably due to animal immigrations coming from deeper soil layers, and that invertebrates would use the cupules more frequently in dry than in wet plots, as these woody structures may serve as moist places that aid to minimize animal desiccation during dry periods of time (microhabitat selection hypothesis), and (2) if these structures are limiting resources in the leaf litter, more invertebrates overall would be found inside relative to outside the cupules as cupule availability increases. That is, artificially increasing the availability of cupules would prompt those animals in the litter to be more likely found using the cupules than in the litter, demonstrating that cupules are limiting resources. Moreover we predicted that (3) the total abundance of fauna (whether inside or outside the cupules) would be affected by cupule availability, with more animals in patches with higher densities of cupules. This is because these structures may enhance spatial heterogeneity, increasing the number of feeding resources, moist micro-patches or hiding places for predator avoidance, i.e., niche space expansion. As a result, we postulated a facilitative effect–particularly in extreme conditions such as droughts–of deciduous trees on soil invertebrates, which in turn actively contribute to leaf litter decay and nutrient turnover. Finally, we predicted that (4) because smaller animals are more susceptible to desiccation, cupules would be preferentially used by smaller invertebrates, particularly in drier conditions.

Materials and Methods

Study area

This study was conducted in an acidic deciduous beech forest (Fagus sylvatica), 1060 m.a.s.l., in a locality near Ricabo (Asturias, North-West Spain; 43°05′52.3″N, 05°59′32.3″W), under permit 2011/059163 of the Asturias Government. Beech forests are a well represented deciduous forest in the Cantabrian Mountains (Sobrino et al., 2009), even though they are at the south-westernmost limit of the distribution range for the species and could be likely declining due to decreasing precipitation associated to global warming (Gutiérrez, 1988; Peñuelas & Boada, 2003). Soil pH in this area is ∼4.1, the estimated annual rainfall is ∼1,121 mm (Ninyerola, Pons & Roure, 2005) and mean tree density is 0.32 trees/m2.

Experimental design

Initial setup

We excluded rainfall from each of four 1 × 1 m fenced plots by covering them with 2 × 2 m plastic roofs located ca. 70 cm height (Appendix 1, Fig. S2). This height was adequate to minimize unwanted dew condensing underneath. Fencing was accomplished by setting a 1 × 1 m galvanized iron enclosure 40 cm height which was buried 10 cm deep in the ground to prevent horizontal macrofauna migrations. Plots were covered with a 1.2 mm mesh size screen on top as well, in order to minimize migrations of macrofauna by climbing the fences. Water exclusion started 18 days before the experiment, beginning on April 17th 2014, and the plots remained covered during the entire study, until June 2nd 2014. The four plots were located in flat micro-sites and relatively close to each other to minimize spatial variation in the distribution of the soil fauna (i.e., the entire experiment extended across less than 50 m2 of the forest area). After 18 days of rainfall exclusion we could be confident that we had induced some water stress within the soil community but not a too severe drought that could affect animal survival within the plot (Verdeny-Vilalta, 2013). Moreover, this period of days without rainfall fits well within the natural range for Iberian beech forests (Ninyerola, Pons & Roure, 2005).

Before the beginning of the experiment, we carefully collected all the beech cupules present in the leaf litter layer of each plot and carried them to the laboratory. A total of 2,001 beech cupules were removed from the four plots and each carefully inspected to collect all the macro- and mesofauna found inside (up to 0.5 mm in body length). To minimize the effect of removing the cupules on the abundance of fauna, all invertebrates (in the order of thousands) were then returned to their respective plots. Additionally, we took measures of 70 cupules by means of a caliper as to assess the external dimensions of these structures (22.1 mm ± 3.0 length–from the base to the opening–and 15.7 mm ± 2.3 average diameter; aperture: 8.9 mm ± 2.9 × 4.4 mm ± 2.5–widest and narrowest diameter of the orifice, respectively). We then oven-heated the cupules at 60 °C during 48 h to ensure the absence of living animals inside prior to releasing the cupules back in the field.

Experimental manipulations: water content and availability of beech cupules

To examine the effect of water and beech cupule availability on the spatial distribution of the soil fauna, we experimentally manipulated the relative humidity of the leaf litter and the density of beech cupules in a split-plot design. Two of the four 1 × 1 m plots were randomly assigned to the ‘Wet’ treatment and the other two to the ‘Dry’ treatment. Within each of these plots, we delimitated six regularly distributed 0.25 × 0.25 m subplots, and randomly assigned them to the following treatments: two of them to the ‘High density of cupules’ treatment, another two to the ‘Intermediate density of cupules’ treatment, and the two remaining subplots to the ‘None’ (i.e., no cupules) treatment. Hence, in each 1 × 1 m plot we established two replicates of the three levels of cupule density (a total of 24 subplots for the entire experiment, Fig. 1). The experimental manipulations were initiated after 18 days of rainfall exclusion, combining the following 6 Water × Cupule density treatments: wet-high, wet-intermediate, wet-none, dry-high, dry-intermediate and dry-none. The delimitation of subplots did not prevent the animals to freely move across treatments within each fenced plot; on the contrary, we aimed at allowing this mobility within each isolated 1 × 1 m plot, to test how animals actively chose a given patch depending on the availability of cupules in either wet or dry conditions.

Figure 1 Outline of the experimental design.

Four experimental plots–two DRY and two WET–and the six subplots delimited within each plot–two of them randomly assigned to the HIGH treatment (high density of cupules), another two to the MED treatment (intermediate density of cupules) and the remaining two to the NONE treatment (no single cupule). Numbers define subplot identity and letters the four consecutive ‘temporal blocks’ (see text for details).

We used commercial mineral spring water (FONT DEL REGÀS®, mineral composition: Ca2+ 32.9 mg/l, Mg2+ 4.1 mg/l, K+ 1.6 mg/l, Na+ 13.5 mg/l, HCO3− 129.2 mg/l, Cl− 7.2 mg/l and SO42− 10.2 mg/l). After the experimental drought and during the whole experiment, wet plots were sprayed with 42 liters of water per week (42 mm/ m2), dispensed in three alternate days (that is, 14 liters/day), while dry plots were watered with only one-sixth of the previous amount, 7 liters per week (7 mm/m2), in only two days (3.5 liters/day). As we applied the watering treatments over four weeks, the total amount of water supplied was 168 liters in each wet plot (168 mm/m2) and 28 in each dry one (28 mm/m2), which falls well within the natural variation in rainfall for the area (Ninyerola, Pons & Roure, 2005). Since collection took place in four temporal blocks (see below), the amount of water accumulated in each sample was different (1/4th the overall amount in the first block, 2/4th in the second temporal block, and so on).

To manipulate the density of beech cupules we used the same cupules removed and dried up at the beginning of the experiment. A total of 480 beech cupules were tagged with a white mark to distinguish the experimental cupules from the few ‘natural’ cupules that could have been inadvertently not removed during the initial setup. Mean density in this forest was 249.6 ± 122.7 cupules/m2 (ca. 15.6 ± 7.7 cupules in 0.25 × 0.25 m; N Melguizo-Ruiz, G Jiménez-Navarro, E De Mas, J Pato, S Scheu, DH Wise, AT Austin & J Moya-Laraño, 2012, unpublished data, N = 10 patches), although in masting years, as appeared to be the previous one, the density can reach up to 500.3 ± 46.8 cupules/m2 (31.3 ± 2.9 cupules in 0.25 × 0.25 m, N = 4 patches). To mimic natural variability in cupule density, and to promote a sufficient experimental effect, 45 cupules were introduced in each of the eight ‘High’ subplots (720 cupules/m2), 15 in the eight ‘Intermediate’ subplots (240 cupules/m2), and none in the eight ‘None’ subplots.

Sampling protocol and measured variables

The 24 subplots were examined in four different periods or ‘temporal blocks’ (in four consecutive weeks), the order of which was randomized: in each temporal block, six subplots were sampled, three subplots randomly collected from one ‘Wet’ plot, and another three subplots randomly collected from one ‘Dry’ plot (see Fig. 1). During each temporal block, the leaf litter from the 6 subplots to sample was rapidly collected to avoid undesired migrations of invertebrates among treatments, and temporarily preserved in plastic containers. Immediately after the collection, we separated the 15 or 45 beech cupules from the leaf litter of each subplot, and kept them in hermetic bags until their inspection in the laboratory. This procedure prevented migrations of fauna from, or towards the surrounding leaf litter. Once in the laboratory, beech cupules from the intermediate and high subplots were scrupulously and individually inspected and all the visible invertebrates found inside identified, counted and measured to the nearest 0.5 mm.

Through the entire study we systematically used the same methodology to estimate leaf litter invertebrate abundances and individual body lengths: in order to avoid damaging the animals during our samplings, and to have more reliable and accurate densities of fauna, we did not sift the leaf litter but rather put it in a tray and closely inspected every leaf, identifying, counting and carefully measuring all the invertebrates found. A total of 12,232 invertebrates were classified to broad taxonomic groups: myriapods, spiders, mites, molluscs and springtails were identified to order, annelids, diplurans and beetles to family level, and the remaining arthropods to class or subclass (Fig. 2).

In order to account for the shape of the different taxa, body lengths were transformed into body masses by using taxon-specific equations from the literature (Hódar, 1996; Hódar, 1997; Sabo, Bastow & Power, 2002; McLaughlin, Jonsson & Emmerson, 2010). For enchytraeids we used the equation mass = 0.0039*length2.53 (where mass and length were measured in mg and mm respectively, O Verdeny-Vilalta, C Guzmán, N Melguizo-Ruiz & J Moya-Laraño, 2013, unpublished data) following the same methodology as Hódar (1996); Hódar (1997).

Figure 2 Taxonomic groups of soil fauna examined in this study.

To assess water content of the leaf litter within each experimental plot, we systematically took one sample per week of the leaf litter around the subplots from each plot (a total of 4 samples each week) and estimated water content by weighing each sample before and after drying it at 60 °C for 24 h. Similarly, we weekly weighed the experimental beech cupules examined (4 samples per temporal block as well) before and after drying them to assess the humidity within the cupules in wet and dry plots. Hence, we could quantify water content from both the leaf litter and the beech cupules for each temporal block.

Statistical analyses

All analyses were performed using the open source statistical software R 3.1.2 (R Core Team, 2014). To examine animal preferences for the cupules, faunal abundances found inside the cupules relative to those found outside were analyzed through Generalized Linear Mixed Models (GLMM), using the “glmer” function in the “lme4” package of R (Bates et al., 2015) with a binomial distribution and a “logit” link function, and with both treatments–water and cupule availability–and their interaction as fixed factors. As no cupule could be used by invertebrates in the no cupule treatment, this treatment was not included in the analysis for preferences. In order to control for differences across taxonomic groups in the response, we included ‘Taxonomic group’ as a random factor (e.g., Ehnes, Rall & Brose, 2011). This was done when analyzing the responses on the entire community. In order to control for potential differences in experimental treatment effects on the diverse groups and plots, both treatments were nested in ‘Taxonomic group’. ‘Temporal block’ nested within ‘Plot’ were also included as random factors to tell the model how subplots were temporally and spatially arranged. Groups that were not present in some of the subplots either inside or outside the cupules were substituted by zeroes. For the sake of comparison, for all analyses including ‘Taxonomic group’ as a random factor, we also performed an additional test in which we added up all animals within each subplot, regardless of their taxonomic affiliation and ran a GLMM, this time with ‘Temporal block’ nested within ‘Plot’ but without distinguishing among taxonomic groups.

This binomial comparison (fauna inside vs. fauna outside the cupules) was used as a measure of preference, considering that the more animals are inside relative to outside, the more they are actively choosing to use the cupules rather than staying in the leaf litter. This was necessary to test predictions (1) more cupule use under drier conditions, and (2) cupules are a limiting factor. If a significant effect was found, we then ran similar models for each of the more abundant taxa (i.e., when numbers were sufficiently high to grant enough variability).

To test part of prediction (1) increase in fauna activity with water availability and prediction (3) facilitative effect of cupule density increasing the overall density of invertebrates, the total abundances of fauna within each subplot (regardless of whether they were found inside or outside the cupules) were also analyzed through GLMMs including the same fixed and random factors, but with a Poisson distribution and a “log” link function. Individual case was included as a random factor to avoid overdispersion from the Poisson model (i.e., variance > mean) (Elston et al., 2001). Again both treatments were nested in ‘Taxonomic group’ (with also the groups that were not present in some of the subplots added as zeroes–zero abundance), and ‘Temporal block’ was nested within ‘Plot’. We tested the interaction between treatments to examine whether effects of cupule availability on invertebrate abundances differed across water levels, which would be the case, for instance, if cupules were more important arranging the community during droughts. Again, if a significant effect was found, we ran similar separate models for each of the most abundant taxa.If any treatment combination was significant we performed post-hoc Tukey tests between treatment levels, for which we used the “glht” function in the “multcomp” package of R (Hothorn, Bretz & Westfall, 2008).

Body masses were transformed to logarithms and analyzed through GLMMs, using the “lmer” function within “lme4”. We included the means of log body masses among individuals as replication units for each subplot. A first model included as a categorical fixed factor the variable ‘Inside/Outside cupules’–which denoted whether the animals were collected within a cupule or in the leaf litter–and ‘Plot’ as random factor. In a second model we aimed to check if the differential use of cupules by small and large invertebrates was affected by water availability and therefore tested the interaction between the ‘Inside/Outside cupules’ variable and the ‘Water treatment’, included as fixed factors, and again ‘Plot’ as random. In a GLMM which included only ‘Taxonomic group’ we found that it explained 80% of the variance (Nakagawa & Schielzeth, 2012) in log body mass (not shown). We therefore ran the analysis both without considering taxonomic affiliation and using the residuals of the above model, as to have a measure of body mass independent on the taxonomic group. The results were qualitatively the same and we therefore only present those for which taxonomic group was not controlled for. This latter analysis assumes that a significant effect found for body mass may or may not depend on taxonomic affiliation, while a model controlling for taxonomic group involves a test on whether the effect of predictors on body mass can occur beyond the effects of taxonomy. We predicted that a significant water treatment*microhabitat interaction would occur in the direction of smaller animals using more frequently beech cupules under dry conditions when compared to larger animals. To test this last hypothesis, if a significant interaction was found, we tested for significant differences between the estimates of the two post-hoc contrasts, i.e., that for dry vs. that for wet conditions, by using a Wald test (Clogg, Petkova & Haritou, 1995). We predicted that the contrast comparing body masses inside vs. outside the cupules would be stronger (i.e., steeper) in dry plots. The appropriate contrasts were obtained by means of the function “lsmeans” in package “lmerTest” (Kuznetsova, Brockhoff & Christensen, 2015) which automatically produces the contrasts for interaction terms from a GLMM with normally distributed errors. All effects in models were tested through a type III deviance analysis (marginal effects), comparing a model with the term of interest, with a model without the target term via either likelihood-ratio tests for normally distributed errors or a Wald tests for binomial and Poisson models, for which we used the “car” package (Fox & Weisberg, 2011). Partial effects of significant variables in models were plotted using the R library “effects” (Fox, 2003). Finally, in order to account for increases in type I error rates in multiple tests, when testing several taxonomic groups after significance, the false discovery rate (FDR) adjustment was applied to correct alpha levels (Benjamini & Hochberg, 1995).

In addition, we used non-metric multidimensional scaling (NMDS), a multivariate ordination method which displays the diverse taxonomic groups in two dimensions, so that dissimilar groups are plotted far apart, while highly similar ones are close together in the ordination space. In this analysis we used the same taxonomic groups, but we distinguished two types of springtails, according to their body lengths: small (<2 mm) and large springtails (≥2 mm), since these animals exhibit a large variation in size. When visualizing multidimensional data in only two dimensions, the distance between each pair of objects is not very accurate. The sum of these errors (or imprecision) is called “stress”, and represents a badness-of-fit measure for the graph, according to which higher stress values are undesirable and reflect that the selected number of dimensions is not adequate to represent the entire spread of the data, being a desirable level of stress <0.2 (Kruskal & Wish, 1978). In order to remove the plot effects in the NMDS ordination (i.e., the fact that subplots within a plot are not independent on each other), we ran a GLM including ‘Plot’ nested within ‘Water treatment’ and took the rounded result of adding the residuals of this model to the raw mean abundance as the dependent variable. Finally, we tested for significance of experimental factors in structuring the community by PERMANOVA (Anderson, 2001), including ‘Water’ and ‘Cupule’ treatments and their interaction as explanatory variables. For NMDS we used the function “metaMDS” with the Bray-Curtis index of dissimilarity and for PERMANOVA the function “adonis”, both within the “vegan” package in R (Oksanen et al., 2015). For PERMANOVA, ‘Plot’ was used in the “strata” option to allow for correct permutation, accounting for the nested structure of the split-plot design.

Results

Effect of experimental watering in the soil and cupules

We successfully induced differences in water availability between wet and dry plots (59.5% ± 1.6 vs. 37.5% ± 1.6 respectively, estimates obtained by averaging across the entire duration of the experiment; χ2 = 96.9, df = 1, p < 0.0001), and water content was progressively higher as the experiment progressed (Estimate = 1.79, χ2 = 3.2, df = 1, p = 0.073). The cupules also were gradually wetter with time (Estimate = 3.99; χ2 = 7.4, df = 1, p = 0.006) but, interestingly, in contrast with the high differences between wet and dry plots in terms of water content in the leaf litter, the differences in the relative humidity of the cupules present in wet and dry plots were virtually nonexistent at the end of the experiment (53.5% ± 0.8 in beech cupules from dry plots vs. 54.3% ± 1.6 in those from wet ones, χ2 = 0.2, df = 1, p = 0.663).

Multiplicative effects: water × cupule interaction

The only significant water × cupule interaction was for the total abundance of springtails (χ2 = 8.4, df = 2, p = 0.0152) (Appendix 2, Fig. S3 and Appendix 3, Table S2a). For the overall fauna, we first found a barely significant water × cupule interaction in the model in which Taxonomic group was included as a random factor (χ2 = 6.3, df = 2, p = 0.042). However, the significant interaction disappeared when we added up abundances of the animals in all groups and ran an analysis without considering the taxonomic affiliation (χ2 = 3.4, df = 2, p = 0.176). We therefore considered the additive model as our final model for total abundances. As no further interaction between the treatments was significant, we avoid mentioning each of them one by one in the remaining of the Results text (see Appendix 3 for further details). Total abundances of fauna in the leaf litter (inside and outside the cupules) are shown in Appendix 3, Table S1.

Relative abundance of fauna inside the cupules: micro-habitat selection

Regardless of cupule availability (partial additive effect), animals were 3.1 times relatively more abundant inside than outside the cupules in dry than in wet plots (χ2 = 28.6, df = 1, p < 0.0001) (Fig. 3A), in particular springtails (χ2 = 84.1, df = 1, p < 0.0001) (Appendix 2, Fig. S4a), mites (χ2 = 43.5, df = 1, p < 0.0001) (Appendix 2, Fig. S5a) and enchytraeids (χ2 = 6.5, df = 1, p = 0.0108) (Appendix 2, Fig. S6a). Regardless of water availability, fauna was also 2.8 times more frequent inside than outside the cupules in patches with higher densities of cupules than in those with intermediate ones (χ2 = 82.0, df = 1, p < 0.0001) (Fig. 3B), and the same groups that responded to water were affected by cupule density; specifically mites (χ2 = 69.5, df = 1, p < 0.0001) (Appendix 2, Fig. S5b), springtails (χ2 = 56.9, df = 1, p < 0.0001) (Appendix 2, Fig. S4b) and enchytraeids (χ2 = 33.4, df = 1, p < 0.0001) (Appendix 2, Fig. S6b).

Figure 3 Abundance of soil fauna inside the cupules relative to outside.

(A) In the water treatment. (B) In the cupule treatment. Abbreviations: DRY, ‘Dry’ treatment; WET, ‘Wet’ treatment. HIGH, ‘High density of cupules’ and MEDIUM, ‘Intermediate density of cupules’ treatment. Effects are model predicted effects ± SE (library “effects”—Fox, 2003). Letters denote significant differences between treatments.

Total abundance of fauna in the leaf litter (whether inside or outside the cupules)

Animals were 1.5 times more frequent in wet than in dry plots (χ2 = 15.2, df = 1, p < 0.0001) (Fig. 4A); this trend was significant for pulmonates –snails and slugs–(χ2 = 33.8, df = 1, p < 0.0001) (Appendix 2, Fig. S7a), enchytraeids (χ2 = 17.4, df = 1, p < 0.0001) (Appendix 2, Fig. S8a), mites (χ2 = 8.4, df = 1, p = 0.0039) (Appendix 2, Fig. S9a), and only marginally significant for lumbricids (χ2 = 5.8, df = 1, p = 0.0157), opilionids (χ2 = 3.5, df = 2, p = 0.0611), lithobiomorphs (χ2 = 2.8, df = 1, p = 0.0953), campodeids (χ2 = 2.8, df = 1, p = 0.0971) and hexapod larvae (χ2 = 2.7, df = 1, p = 0.0991). Moreover, the total abundance of fauna increased with cupule availability (χ2 = 42.3, df = 2, p < 0.0001) (Fig. 4B); animals were 1.4 and 1.5 times more abundant in patches with higher densities of cupules than in patches with intermediate densities of cupules (Tukey contrasts, p < 0.0001) and without cupules at all (Tukey contrasts, p < 0.0001), respectively. This increasing abundance with greater cupule availability was significant for mites (χ2 = 250.4, df = 2, p < 0.0001) (Appendix 2, Fig. S9b), enchytraeids (χ2 = 23.4, df = 2, p < 0.0001) (Appendix 2, Fig. S8b), hexapod larvae (χ2 = 20.7, df = 2, p < 0.0001), and marginally pulmonates (χ2 = 5.8, df = 2, p = 0.0541) (Appendix 2, Fig. S7b) and pseudoscorpionids (χ2 = 5.0, df = 2, p = 0.0808). On the contrary, campodeids were more abundant with decreasing densities of cupules (χ2 = 12.4, df = 2, p = 0.0020), and lithobiomorphs were less abundant in patches with intermediate densities of cupules than in those with high cupule density and without cupules (χ2 = 10.3, df = 2, p = 0.0059) (Appendix 3, Table S2b).

Figure 4 Total abundance of soil fauna in the leaf litter (inside and outside the cupules).

(A) In the water treatment. (B) In the cupule treatment. Abbreviations: DRY, ‘Dry’ treatment; WET, ‘Wet’ treatment. HIGH, ‘High density of cupules’, MEDIUM, ‘Intermediate density of cupules’ and NONE: ‘None’ (i.e., no single cupule) treatment. Effects are model predicted effects ± SE (library “effects”—Fox, 2003). Letters denote significant differences between treatments after post-hoc contrasts.

Body masses and cupule use

Animals found inside the cupules were significantly smaller than those found outside (χ2 = 163.4, df = 1, p < 0.0001). Furthermore, in dry plots smaller animals seemed to use the cupules in a greater degree than in wet plots, as we found a stronger effect in dry relative to wet plots when comparing the mass of animals inside cupules vs. those outside (interaction Inside/Outside × Water treatment, χ2 = 25.1, df = 1, p < 0.0001; post-hoc contrast in dry plots, estimate ± SE = −1.7 ± 0.1; post-hoc contrast in wet plots, 0.9 ± 0.1; comparison between estimates, Z = 5.06, p < 0.0001; Fig. 5).

Figure 5 Interactive effect of water availability on the log-body mass (mg) of soil invertebrates found inside and outside the beech cupules.

Abbreviations: DRY, ‘Dry’ treatment; WET, ‘Wet’ treatment. INSIDE CUPULES, found inside cupule; OUTSIDE CUPULES, found in the surrounding leaf litter. Effects are model predicted effects ± SE (library “effects”—Fox, 2003). Animals in cupules were smaller than animals outside and the effect was stronger in DRY than in WET plots (see text for details on statistical analysis).

Effects of treatments on community structure

The results of PERMANOVA showed that both water (pseudo-F = 13.2, p = 0.0001, partial-R2 = 0.246) and cupule treatments (pseudo-F = 10.4, p < 0.0001, partial-R2 = 0.388) significantly affected community structure additively, as the interaction was not significant (pseudo-F = 0.75; p = 0.521; partial-R2 = 0.028). For the NMDS, the stress in two dimensions for the output shown in Fig. 6 was 0.052, indicating an adequate fit to the data. Smaller animals such as mites, enchytraeids, pseudoscorpions and small springtails were ordered close to subplots with higher densities of cupules, while larger animals–spiders, centipedes, campodeids and large springtails–were ordered far apart from these patches, nearer the subplots without cupules. Secondary decomposers, mesostigmatid mites and slugs ordered around wet subplots, whereas other small predators (prostigmatid mites, pseudoscorpions) ordered close to drier ones.

Figure 6 Non-metric multidimensional scaling (NMDS).

Discussion

In general the interaction between water and cupule treatments was not significant, although we found an important trend for the total abundance of springtails, whose positive response to higher cupule densities was stronger in dry plots. This result suggests that these animals are closely related to beech cupules, and that this relation is influenced by water availability (i.e., soil moisture). We found important additive effects of both experimental treatments on faunal distributions as well. First, there were greater abundances of fauna inside the cupules relative to outside in dry than in wet plots, and in patches with higher cupule densities than in patches with intermediate densities or without cupules. Second, total numbers of animals–inside and outside the cupules–were higher in wet than in dry plots, and also increased with cupule availability. In addition, there were significant differences in body mass in animals found inside and outside the cupules, with a greater use of these structures by smaller invertebrates. This trend was stronger in dry plots, as predicted if cupules are used by smaller animals to avoid desiccation.

Beech cupules as ‘moist shelters’, water availability and spatial heterogeneity

The finding that the abundance of fauna collected inside the cupules relative to outside was higher in dry than in wet plots suggests that soil animals–particularly springtails, mites and enchytraeids–use the cupules more in drier conditions. To deal with water scarcity and low humidity levels, these animals may have to shelter and seek for moist micro-patches (Wolters & Ekschmitt, 1997), either in deeper leaf litter layers or in superficial cavities and crevices. These woody husks do have the capacity to hold moisture in a greater degree and for longer than does leaf litter, and therefore can effectively act as ‘moist shelters’ during drought periods. Furthermore, invertebrates inside the cupules were smaller than those found in the leaf litter, pointing out that the use of these structures can be limited by body mass. Although most macrofauna should be able to go into the cupules, the tight space inside may complicate their use by large-bodied animals. Additionally, this effect on body mass was stronger in dry than in wet plots; the greater use of cupules by smaller animals is more patent in drier conditions, indicating that these husks are more important for invertebrates with smaller body masses, especially under droughts. It is well known that larger animals resist desiccation better than smaller ones, owing to their lower surface to volume ratio, which allow them to hold the water for longer time in their bodies (Chown, 1993; Le Lagadec, Chown & Scholtz, 1998; Renault & Coray, 2004). Therefore, smaller animals make more use of cupules during droughts probably because they are more prompt to desiccation.

The total abundance of fauna was higher in wet than in dry plots, indicating that rainfall and the subsequent soil moisture favour the immigration of soil animals from the deeper layers into the surface, chiefly enchytraeids, mites and pulmonates, and marginally lumbricids, lithobiomorphs, opilionids, campodeids and hexapod larvae. As found in previous studies, these results stress that water availability is a decisive factor determining the spatial distribution of soil animals (Levings & Windsor, 1984; Grear & Schmitz, 2005; Melguizo-Ruiz et al., 2012) and the configuration of soil food-webs, not only in extreme water-limited ecosystems, but also in deciduous forests (Verdeny-Vilalta, 2013; Verdeny-Vilalta & Moya-Laraño, 2014). Summarizing, soil moisture determines the distribution and abundance of soil fauna, and beech cupules act as moist shelters that may reduce invertebrate desiccation and hence facilitate their maintenance during drought periods.

The number of invertebrates found inside the cupules relative to those found outside increased with cupule density, indicating that most invertebrates such as mites, springtails and enchytraeids may be attracted to these structures based on their availability and hence, that these structures are limiting resources in the forest floor. Moreover, regardless of whether they were found inside or outside the cupules, the overall number of invertebrates was also higher in patches with higher densities of cupules relatively to patches with intermediate densities or without cupules at all. The positive effect of cupule availability was evident for the most abundant groups in the leaf litter (mites, enchytraeids, hexapod larvae and, marginally, pulmonates and pseudoscorpionids), although some animals exhibited different and even opposite patterns (for instance, lithobiomorphs and campodeids). Beyond body size or even shape, the fact that some taxa avoided areas with high availability of cupules may have to do with their preference for resources associated with the litter itself, or even for micro-patches that are physically less holey than those rich in cupules. Nevertheless, the presence of cupules may increase the spatial heterogeneity of the habitat and strongly influence the micro-distribution of fauna.

The positive effect of cupule density on both the total abundance of fauna in the leaf litter and the total abundance of fauna found inside the cupules suggests that these fruit husks are valuable and limiting resources for most soil animals. Indeed, beech cupules could serve to reduce encounters with predators, to feed on microorganisms growing inside, as ‘moist shelters’ or to satisfy any other motivation. Note that, despite the effects were additive in most cases, the interaction water × cupule availability was significant for the total abundance of springtails. The multiplicative effect of both water and cupule treatments reflects a positive effect of cupule density on their abundance stronger in dry plots, indicating that for these animals the cupules are more important limiting resources when water is scarce. This result could support the ‘stress-gradient hypothesis’, according to which under severe conditions or stress (any environmental factor having detrimental effects on organisms when exceeding a given threshold, such as heat or desiccation), positive or facilitative effects are more relevant or decisive (Bertness & Callaway, 1994; Bertness et al., 1999; Stachowicz, 2001; Callaway et al., 2002; Kawai & Tokeshi, 2007; Anthelme et al., 2012).

Community spatial arrangement

Our analysis of the community structure (Ordination using NMDS followed by PERMANOVA) shows that the soil community is not randomly distributed in space, but strongly structured around cupule and water availability. As found in the analyses of abundances and body sizes, some groups (especially small animals) appeared to more closely depend on the cupules than others; Oribatida, Mesostigmata and Prostigmata mites, small springtails, pseudoscorpionids and enchytraeids were strongly related to the cupules, whereas other, relatively larger animals such as centipedes, spiders and large springtails, were rarely found near or inside these structures. As commented above, even the larger invertebrates may be able to get into the cupules, but the minute space available inside may probably discourage them from using such husks as a frequent refuge. Besides, some groups appeared to seek for wetter patches while others did not, likely reflecting differences in the responses of soil organisms with respect to soil moisture. This study evidences that the distribution of soil fauna in the forest floor is greatly affected by cupule and water availability, which may in turn likely influence the structure and dynamics of the soil food web and associated ecosystem processes, such as litter decomposition.

Beech cupules as keystone structures

According to our experimental results, beech cupules—and probably other fruit husks—should be considered ‘keystone structures’ (as defined in Tews et al., 2004), as they appear to provide refuge, may increase spatial heterogeneity, and presumably serve as resources to diverse groups of species, especially abundant in forest soils. Although we only estimated faunal abundances rather than diversity, and therefore we did not demonstrate that cupules increased species richness in the leaf litter, we showed that these structures widely favour several groups of decomposers and some predatory animals, and thus contribute to the maintenance of the soil food web.

Interestingly, it has been shown that rainfall negatively correlates with fruit production in oaks, probably because wet conditions limit pollen flow and fertilization (Knops, Koenig & Carmen, 2007). Similarly, beech cupule abundance is negatively related to the mean annual precipitation in northern Spain (N Melguizo-Ruiz & J Moya-Laraño, 2013, unpublished data), denoting that the input of husks to the soil is higher in drier sites. Springtails appeared to be more affected by cupule availability in drier conditions, agreeing with the ‘stress-gradient hypothesis’. Since fruit crop is higher in drier areas, and this group seemed to depend more strongly on cupules during drought periods, the indirect and facilitative effect of trees on these populations may be more relevant in dry than in rainy forests, contributing to their maintenance. In our study, beech trees not only widen the range of conditions where soil organisms can live through these cupules and facilitate their maintenance during droughts, but also increase litter heterogeneity and enhance faunal abundances under mild conditions as well. Beech cupules could actually prevent vertical migrations of the fauna to deeper micro-sites during droughts, in which case trees would contribute to the stability and maintenance of soil animal populations in the leaf litter, and hence potentially increase the efficiency in nutrient recycling below their canopies. Beech trees could therefore be considered ‘autogenic engineers’ as defined by Jones, Lawton & Shachak (1994), since they modify the environment–here, the leaf litter–through their own physical structures–their fruit husks–, and consequently exert an indirect facilitative effect that benefits soil invertebrates during stressful conditions such as droughts.

Finally, when both members are favoured by an interaction, such relation is considered mutualistic (Bronstein, 1994; Stachowicz, 2001). Given the potential contribution to mitigate desiccation by these fruit husks, expanding the micro-habitat range for many soil invertebrates which are actually responsible for ensuring future nutrient availability to long-leaving trees (Edwards, Reichle & Crossley Jr, 1970; Lavelle, 1997; Huhta, 2007), a potential mutualistic interaction between deciduous trees and the decomposer community may emerge as an important driver of forest soil regulation and maintenance.

Conclusions

We have demonstrated how beeches can indirectly enhance the maintenance of soil biota during unfavourable conditions such as droughts through their woody fruit husks. Beech cupules, which presumably contribute to increase spatial heterogeneity and seem to be limiting and valuable resources to the most abundant groups of soil invertebrates, especially the smaller ones, appear to provide moisture sheltering in dry conditions. These husks should therefore be considered as ‘keystone structures’ that contribute to the maintenance of the soil community, and likely indirectly, to its associated ecosystem function, leaf litter decomposition.

Supplemental Information

Supplemental Information 1 Supplementary Material

Appendix 1, 2 and 3 (Beech cupules as keystone structures for soil fauna)

Click here for additional data file.

Supplemental Information 2 Data - Experiment Beech Cupules

Click here for additional data file.

We thank E Palop for helping in the field. We are also grateful to all who contributed with their advises to the experimental design.

Additional Information and Declarations

Competing Interests

Author Contributions

Field Study Permissions

Data Availability

The authors declare there are no competing interests.

Nereida Melguizo-Ruiz conceived and designed the experiments, performed the experiments, analyzed the data, wrote the paper, prepared figures and/or tables, reviewed drafts of the paper.

Gerardo Jiménez-Navarro conceived and designed the experiments, performed the experiments, wrote the paper, prepared figures and/or tables, reviewed drafts of the paper.

Jordi Moya-Laraño conceived and designed the experiments, analyzed the data, wrote the paper, prepared figures and/or tables, reviewed drafts of the paper.

The following information was supplied relating to field study approvals (i.e., approving body and any reference numbers):

Asturias Government, permit 2011/059163.

The following information was supplied regarding data availability:

The raw data has been supplied as Supplementary File.

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
