# Peer review of "Beech cupules as keystone structures for soil fauna"

_PeerJ, doi:10.7717/peerj.2562_

## Round 0.1 · original submission · Major Revisions

You'll see that there are a number of small points and a few big points. In particular, pay close attention to Reviewer 1's comments about your analysis. If you cannot deal with his/her comments, I think it will be tough to consider this paper further. In addition - addressing the comments by all 3 reviewers will dramatically improve the paper.

Reviewer 1 ·

Basic reporting

This is a very interesting paper and the introduction does a good job of providing sufficient background on facilitation and keystone structures and highlighting the research gaps that will be addressed.

The figures are relevant, but it would be more informative to see the raw data and SEs in the figures, rather than the predicted values. I think that presenting the data this way obscures some of the variation (the SEs are extremely low for an N of 2 (table in appendix 3)) and also does not allow the paper to be readily used in meta-analyses.

The writing is clear but some sentences could be shortened/simplified (e.g. line 93-96; line 113 – start this sentence with “low relative…”). Try to restructure the sentences (e.g. divide into two sentences) on line 78 and line 86 to remove the sections in parentheses.
Some word choices should be changed:
line 77 – lapse does not seem like the best word to use here
Line 105, the use of the word ‘first’ seems unnecessary since there is no second
Line 117 – change loss to lose
Line 194 – would be clearer to say None (i.e., no cupules)
Line 212 being = been
Line 232 – should be nonexistent not inexistent
Line 499 – “we showed” instead of we actually proved

The hypotheses could be reorganized so that both of the moisture related hypotheses are together (1 and 3).

Line 414/415 – could delete the last sentence in this paragraph

Experimental design

The research question is clearly identified and methods are generally well described.
-How far apart were the plots (line 169)?
-Why was commercial mineral water used? How does this compare to rain water?
-When were the different temporal blocks sampled (i.e., what was the timing of the sampling?) It looks like they were a couple of weeks apart according to the raw data?
-On line 251, was it 122 321 invertebrates? (I assume the decimal point should not be there)

Validity of the findings

The level of replication does not seem adequate (2 dry and 2 wet plots) to obtain a very reliable estimate of variance, especially with two fixed effects, an interaction, and three random effects in some of the models. I do not think the article meets PeerJ's standards in that respect. For random effects, there is some argument that there should be enough levels of the factor in the data that an estimate of the variance of the population of effects can be based on them which I don't think is the case here. I’m not sure about the use of taxonomic group as a random effect either – can some further explanation of this be added? What is the ‘case’ variable that was used as an additional random factor?

In addition, the subplots were sampled at different time periods but this does not appear to have been accounted for in the models. Or is the case variable actually for different time periods?

What distance metric was used for the NMDS and how many iterations were run?

Is it possible that abundance of fauna was lower in dry plots due to reduced survival, rather than immigration to deeper layers?

Reviewer 2 ·

Basic reporting

I found this manuscript to be interesting and nicely written. I do not have much to say about its basics. The language is good and clear. The length of the different sections is adequate. I like it very much you have an experiment. The database is superb (the amount of work you should have done sorting and counting all those litter critters is chilling). The stats are appropriate, clear, and accompanied by nice figures. The discussion is well framed (but see my comments below).

Experimental design

Excellent! But see my comments below.

Validity of the findings

See my comments below.

Additional comments

Like I said before, I liked it very much to read your manuscript. However, there are some major issues that I would like you to review.

First, I think that the theoretical background is weak. It seems you framed your manuscripts using so many *important* concepts, i.e. facilitation, ecosystem engineers, niche construction and keystone structures, that it gave me the feeling that you tried way too much to "hype" it. For example, I wonder if you really are dealing with a case of facilitation (by the way, not including any of the manuscripts by Maestre et al. --the group with most recent and important research in the topic-- is weird). It is my understanding that, and I can easily see how, a plant facilitates the living of other plant in an almost permanent way, by reducing stress in some of its forms. But in your case, I have many problems considering beech cupules as something different to resource provisioning. My main point is that beech cupules are a *dead* substrate. And soil food webs rely, by definition, in dead substrates. Beech cupules are not serving soil critters in a different way (and maybe less so) than dead wood is serving sacroxylic fauna! Right? I may be loosing something, but I really find it difficult to frame your manuscript as a case of facilitation. I find similar issues with the other concepts you invoke. Are (all) plant species that get rid of dead plant tissues ecosystem engineers? If they are building niches, beech cupules need to have species specialized (in some form) to them. I am not sure you are able to test this idea with your dataset. I find it difficult for any taxa to specialize in something dead.

Second. I also feel uneasy with the concept of beech cupules as keystone structures. You did a superb work, showing with a great experiment that soil fauna benefit from beech cupules, specially so when it gets dry. I clearly understand the idea of separating the effects of one/two factor(s) experimentally (in your case beech cupules and water). But the litter is full of other structures and components that you did not include in your experiment. For example, likely, a lot of the soil fauna dig themselves deep in the soil during dry periods of time. Is the *soil* another keystone structure? More important, of the total soil fauna what % hide themselves in the beech cupules vs. hide in the soil? My point is that in your manuscript you clearly showed that the smaller soil fauna uses beech cupules as shelter, and they do more so in dry times, but you have not tested for fauna preferences on other substrates (not sure if you in Europe have acorns, for example) and how preference for beech cupules differs from preference to other substrates. I am neither suggesting that you need to do these other experiments nor saying that your work lacks value, I just want to see how other litter compartments affect litter critters before I say that beech cupules are a major part of soil biology.

That is all I have to say about your manuscript. Again, I suggest you to revise many of these underlying concepts and use them more conservatively across your manuscript. Because dealing with these suggestions (if you consider my comments appropriate) will likely change profoundly the theoretical framework of your work, which subsequently will affect all other parts of the manuscript, I suggest to the editor your manuscript to be rejected.

·

Basic reporting

No comments

Experimental design

No comments

Validity of the findings

See general comments for the author

Additional comments

This is a nice experimental study on the effect of beech cupules on soil fauna in Spain. The study combines a desiccation treatment, with an experimental gradient in beech cupule numbers and finds interesting responses in the soil fauna. The study is generally well presented and certainly valid for publication, but revision, especially on structural aspects are needed before publication.

Major points:
1) The order in which the hypotheses are posed could be improved. I suggest to start with the simpler hypotheses, on single factors, i.e. overall effects of number of beech cupules and water treatment on fauna, before going to the inside/outside use of beech capsules and the interactions between the two treatments. This restructuring should be repeated also for the Results section.
2) The Material and Methods section is very wordy, especially in relation to statistical analysis. Try to condensate, and move results on beech capsule humidity (line 220-233) to the Results
3) Please provide one or mode tables to give summary results of the main model runs. The many statistics given in the main text makes part of the Results section very heavy to read

Minor points:
53: members -> partners
88-93: Rewrite to improve clarity. The cross-pollination aspect seems out of context
103: Both “seed husk”, “fruit husk” and “cupule” are used throughout the MS. One of the two former expressions should be used throughout at least as a more general expression (across tree genera), whereas cupules can be used specifically for beech, as a special case
106: fungivorous
107: Rewrite to improve clarity
117: loss -> lose
132-33: To me this hypothesis is illogical. I would expect higher use of a limiting resource under low resource availability in harsh conditions rather than the opposite.
134: posited -> hypothesized
140-142: I don’t think this hypothesis makes sense. You are studying a purely (?) deciduous forest ecosystem, and hence it is absurd to think that deciduous trees, as formers of the system, are not important for the processes going on in that system. The hypothesis would be valid if you studied e.g. the role of single intermixed beech trees in coniferous dominated systems, but this is not the case here.
151-152: Please provide more details on the study system, e.g. stand age, current/past management admixing tree species, dominant herbs, soil pH, humus type. The study system is indicated to be primary forest, but is this really true considering common interpretations of this concept, e.g. https://en.wikipedia.org/wiki/Old-growth_forest ?
175-176: Clarify at what stage of the experiment this was done. Also clarify how deep down to the litter layer you searched for cupules (i.e. only in the litter layer?)
195-196: Better to start this part by “The experimental manipulations were initiated after the 18 days of rainfall…”
202-209: Even if this can be easily recalculated to mm of rainfall, please provide the measures also on this scale to help the reader
212: being -> been
214-5: Unclear what “N” refers to
215-6: Rewrite to clarify text. I guess you refer to the preceding year, rather than the actual study year, as a mast year?
216-7: Better: “To mimic natural variability in cupule density, and to promote…”
218-9. Please provide these densities also as cupules/m2
220-233. Give more details on 1) when sampling was done, 2) how many subsamples were taken, 3) and for how many days the sampling continued. In addition give the results on the measured differences in water content etc. in the Results section (see also major point 2).
240: delete “so as”
252. delete “then”
302-3: Clarify text in the parenthesis
341-3: It seems a bit confusing that the results start out with some of the more complex results, especially since they are not really clear. See also my major point 1.
350-79: A lot of test results are given, with full details, making the text hard to read. Please provide main results in a table, and focus the text to guide the reader to understand these tables correctly. E.g. all groups, except XXX showed a positive response to XXX (Table YY)-
391-3: I guess these results stems from the Permanova? If yes indicate this in the text. Otherwise remember to provide the Permanova results
400-1: The sentence “probably because water stimulated the activity and immigration differently for different taxa” can be deleted.
406-7: The opening of the discussion should be more specific, and explicitly mention the two treatments
420. Delete “likely”
423-6: This sound like a repetition of the Introduction. But now you have data to validate this expectation (wrongly presented in the Material and methods section, lines 227-230). Rewrite your text to reflect this.
442-3: be more specific on “water-limited ecosystems”. Your results clearly indicate that deciduous forests, in some aspects, can be considered water-limited.
445-5. Don’t generalize too much here. You only studied beech cupules.
447: delete “in addition”
447-56: Again, I wouldn’t expect this result (cf. my comments to lines 132-33). I am no expert in this field, but please check how your results match similar studies on habitat use under variable stress/habitat abundance regimes. Your results at least indicate that even the high cupule treatment is below the limit where the number of cupules stops to be limiting. Was this what you intended and how does this compare to natural variation in this habitat?
463-5. Please rewrite
484: “springtails, were”
497-501: Please rewrite
514: in which case trees would -> and hence
520-9: I have sympathy with this interesting and bold interpretation, but it need to be given in a shorter way, as a hypothesis for further testing, and without repeating the definition of mutualism. It is quite likely that these short term desiccation effects have no effect on the overall year-round soil processes, provided that sufficient water is present in long enough periods. There is even evidence that microclimatic fluctuations may in fact enhance soil biodiversity and decomposer processes, e.g: http://www.sciencedirect.com/science/article/pii/S0031405604701080 and http://link.springer.com/article/10.1007/s00442-006-0406-3
Fig. 1 and perhaps even fig. 2 could be easily moved to the appendix without losing clarity.

---

## Round 0.2 · accepted · Accept

Nice work on the revision. You took the comments seriously, and your work has improved the manuscript considerably.